# Contextual Squeeze-and-Excitation for Efficient Few-Shot Image Classification

**Massimiliano Patacchiola**
University of Cambridge
mp2008@cam.ac.uk

**John Bronskill**
University of Cambridge
jfb54@cam.ac.uk

**Aliaksandra Shysheya**
University of Cambridge
as2975@cam.ac.uk

**Katja Hofmann**
Microsoft Research
kahofman@microsoft.com

**Sebastian Nowozin***
nowozin@gmail.com

**Richard E. Turner**
University of Cambridge
ret26@cam.ac.uk

## Abstract

Recent years have seen a growth in user-centric applications that require effective knowledge transfer across tasks in the low-data regime. An example is personalization, where a pretrained system is adapted by learning on small amounts of labeled data belonging to a specific user. This setting requires high accuracy under low computational complexity, therefore the Pareto frontier of accuracy vs. adaptation cost plays a crucial role. In this paper we push this Pareto frontier in the few-shot image classification setting with a key contribution: a new adaptive block called Contextual Squeeze-and-Excitation (CaSE) that adjusts a pretrained neural network on a new task to significantly improve performance with a single forward pass of the user data (context). We use meta-trained CaSE blocks to conditionally adapt the body of a network and a fine-tuning routine to adapt a linear head, defining a method called UpperCaSE. UpperCaSE achieves a new state-of-the-art accuracy relative to meta-learners on the 26 datasets of VTAB+MD and on a challenging real-world personalization benchmark (ORBIT), narrowing the gap with leading fine-tuning methods with the benefit of orders of magnitude lower adaptation cost.

## 1 Introduction

In recent years, the growth of industrial applications based on recommendation systems (Bennett et al., 2007), speech recognition (Xiong et al., 2018), and personalization (Massiceti et al., 2021) has sparked an interest in machine learning techniques that are able to adapt a model on small amounts of data belonging to a specific user. A key factor in many of these applications is the Pareto frontier of accuracy vs. computational complexity (cost to adapt). For example, in a real-time classification task on a phone, a pretrained model must be personalized by exploiting small amounts of data on the user's device (context). In these applications the goal is twofold: maximize the classification accuracy on unseen data (target) while avoiding any latency and excessive use of computational resources.

Methods developed to face these challenges in the few-shot classification setting can be grouped in two categories: meta-learning and fine-tuning. Meta-learning is based on the idea of learning-how-to-learn by improving the algorithm itself (Schmidhuber, 1987; Hospedales et al., 2020). Meta-learners are trained across multiple tasks to ingest a labeled context set, adapt the model, and predict the class membership of an unlabeled target point. Fine-tuning methods adjust the parameters of a pretrained neural network on the task at hand by iterative gradient-updates (Chen et al., 2019; Triantafillou et al., 2019; Tian et al., 2020; Kolesnikov et al., 2020; Dumoulin et al., 2021).

---

*Work done while the author was at Microsoft Research – Cambridge (UK)

36th Conference on Neural Information Processing Systems (NeurIPS 2022).

We can gain an insight on the differences between those two paradigms by comparing them in terms of accuracy and adaptation cost. Figure 1 illustrates this comparison by showing on the vertical axis the average classification accuracy on the 18 datasets of the Visual Task Adaptation Benchmark (VTAB, Dumoulin et al. 2021), and on the horizontal axis the adaptation cost measured as the number of multiply–accumulate operations (MACs) required to adapt on a single task (see Appendix C.1 for details). Overall, fine-tuners achieve a higher classification accuracy than meta-learners but are more expensive to adapt. The comparison between two state-of-the-art methods for both categories, Big Transfer (BiT, Kolesnikov et al. 2020) and LITE (Bronskill et al., 2021), shows a substantial performance gap of $14\%$ in favor of the fine-tuner but at a much higher adaptation cost, with BiT requiring $526 \times 10^{12}$ MACs and LITE only $0.2 \times 10^{12}$ MACs.

It is crucial to find solutions that retain the best of both worlds: the accuracy of fine-tuners and low adaptation cost of meta-learners. The main bottleneck that hampers the adaptation of fine-tuners is the need for multiple gradient adjustments over the entire set of network parameters. Restricting those adjustments to the last linear layer (head) significantly speeds up fine-tuning, but it harms performance (e.g. see experiments in Section 5.1). Finding a way to rapidly adapt the feature extractor (body) is therefore the main obstacle to bypass. In this paper we propose a hybrid solution to this issue, exploiting meta-learned adapters for rapidly adjusting the body and a fine-tuning routine for optimizing the head.

At the core of our approach is a novel extension of the popular Squeeze-and-Excitation block proposed by Hu et al. (2018) to the meta-learning setting that we call **C**ontextu**a**l **S**queeze-and-**E**xcitation (**CaSE**). We exploit CaSE as building block of a hybrid training protocol called **UpperCaSE** which is based on the idea of adjusting the body of the network in a single forward pass over the context, and reserving the use of expensive fine-tuning routines for the linear head, similarly to methods like MetaOptNet (Lee et al., 2019), R2D2 (Bertinetto et al., 2018), and ANIL (Raghu et al., 2019). Figure 1 shows how UpperCaSE substantially improves the performance

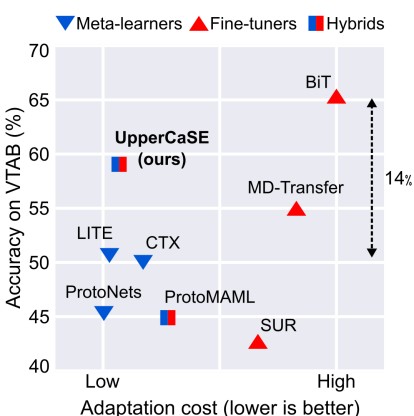

Figure 1: Accuracy and adaptation cost on VTAB for meta-learners (blue), fine-tuners (red), and hybrids (blue-red). Black dotted-line is the previous Pareto front across categories. UpperCaSE narrows the gap with the leading fine-tuning method and represents the best trade-off in terms of accuracy/adaptation-cost.

in the low-cost regime, outperforming meta-learners, fine-tuners such as MD-Transfer (Triantafillou et al., 2019), and reducing the gap with the current state of the art (BiT). When adaptation cost is critical, UpperCaSE is the best method currently available since it can provide substantial computation savings and compelling classification performance.

Our *contributions* can be summarized as follows:

1. We introduce a new adapter called **C**ontextu**a**l **S**queeze-and-**E**xcitation (**CaSE**), based on the popular Squeeze-and-Excitation model proposed by Hu et al. (2018), that outperforms other adaptation mechanisms (e.g. the FiLM generators used in Bronskill et al. 2021) in terms of parameter efficiency (a $75\%$ reduction in the number of adaptation parameters) and classification accuracy (a $1.5\%$ improvement on MetaDataset and VTAB). The code is released with an open-source license [1].

2. We use CaSE adaptive blocks in conjuction with a fine-tuning routine for the linear head in a model called UpperCaSE, reporting an improved classification accuracy compared to the SOTA meta-learner (Bronskill et al., 2021) on the 8 datasets of MDv2 ($+2.5\%$ on average) and the 18 datasets of VTAB ($+6.8\%$ on average), narrowing the gap with BiT (Kolesnikov et al., 2020) with the benefit of orders of magnitude lower adaptation cost.

3. We showcase the potential of UpperCaSE in a real-world personalization task on the ORBIT dataset (Massiceti et al., 2021), where it compares favorably with the leading methods in the challenging cross-domain setting (training on MDv2, testing on ORBIT).

---

[1] https://github.com/mpatacchiola/contextual-squeeze-and-excitation

## 2 Contextual Squeeze-and-Excitation (CaSE)

**Problem formulation** In this paragraph we introduce the few-shot learning notation, as this will be used to describe the functioning of a CaSE adaptive block. Let us define a collection of meta-training tasks as $\mathcal{D} = \{\tau_1, \ldots, \tau_D\}$ where $\tau_i = (\mathcal{C}_i, \mathcal{T}_i)$ represents a generic task composed of a context set $\mathcal{C}_i = \{(\mathbf{x}, y)_1, \ldots, (\mathbf{x}, y)_M\}$ and a target set $\mathcal{T}_i = \{(\mathbf{x}, y)_1, \ldots, (\mathbf{x}, y)_D\}$ of input-output pairs. Following common practice we use the term *shot* to identify the number of samples per class (e.g. 5-shot is 5 samples per class) and the term *way* to identify the number of classes (e.g. 10-way is 10 classes per task). Given an evaluation task $\tau_* = \{\mathcal{C}_*, \mathbf{x}_*\}$ the goal is to predict the true label $y_*$ of the unlabeled target point $\mathbf{x}_*$ conditioned on the context set $\mathcal{C}_*$.

In fine-tuning methods, we are given a neural network $f_{\boldsymbol{\theta}}(\cdot)$, with parameters $\boldsymbol{\theta}$ estimated via standard supervised-learning on a large labeled dataset (e.g. ImageNet). Given a test task $\tau_*$ adaptation consists of minimizing the loss $\mathcal{L}(\cdot)$ via gradient updates to find the task-specific parameters $\boldsymbol{\theta}_{\tau_*} \leftarrow G(\epsilon, \mathcal{L}, \tau_*, f_{\boldsymbol{\theta}})$, where $\epsilon$ is a learning rate, and $G(\cdot)$ is a functional representing an iterative routine that returns the adapted parameters $\boldsymbol{\theta}_{\tau_*}$ (used for prediction). This procedure is particularly effective because it can exploit efficient mini-batching, parallelization, and large pretrained models.

In meta-learning methods training and evaluation are performed episodically (Vinyals et al., 2016), with training tasks sampled from a meta-train dataset and evaluation tasks sampled from an unseen meta-test dataset. The distinction in tasks is exploited to define a hierarchy. The parameters are divided in two groups: $\boldsymbol{\phi}$ task-common parameters shared across all tasks (top of the hierarchy), and $\boldsymbol{\psi}_\tau$ task-specific parameters estimated on the task at hand as part of an adaptive mechanism (bottom of the hierarchy). The way $\boldsymbol{\phi}$ and $\boldsymbol{\psi}_\tau$ come into play is method dependent; they can be estimated via gradient updates (e.g. MAML, Finn et al. 2017), learned metrics (e.g. ProtoNets, Snell et al. 2017), or Bayesian methods (Gordon et al., 2018; Patacchiola et al., 2020; Sendera et al., 2021).

**Standard Squeeze-Excite (SE)** We briefly introduce standard SE (Hu et al., 2018), as we are going to build on top of this work. SE is an adaptive layer used in the supervised learning setting to perform instance based channel-wise feature adaptation, which is trained following a supervised protocol together with the parameters of the neural network backbone. Given a convolutional neural network, consider a subset of $L$ layers and associate to each one of them a Multi-Layer Perceptron (MLP), here represented as a function $g_{\boldsymbol{\phi}}(\cdot)$. The number of hidden units in the MLP is defined by the number of inputs divided by a reduction factor. Given a mini-batch of $B$ input images, each convolution produces an output of size $B \times C \times H \times W$ where $C$ is the number of channels, $H$ the height, and $W$ the width of the resulting tensor. For simplicity we split this tensor into sub-tensors that are grouped into a set $\{\mathbf{H}_1, \ldots, \mathbf{H}_B\}$ with $\mathbf{H}_i \in \mathbb{R}^{C \times H \times W}$. To avoid clutter, we suppress the layer indexing when possible. SE perform a spatial pooling that produces a tensor of shape $B \times C \times 1 \times 1$; this can be interpreted as a set of vectors $\{\mathbf{h}_1, \ldots, \mathbf{h}_B\}$ with $\mathbf{h}_i \in \mathbb{R}^C$. For each layer $l$, the set is passed to the associated MLP that will generate an individual scale vector $\boldsymbol{\gamma}_i \in \mathbb{R}^C$, where

$$\boldsymbol{\gamma}_1^{(l)} = g_{\boldsymbol{\phi}}^{(l)}\left(\mathbf{h}_1^{(l)}\right) \quad \cdots \quad \boldsymbol{\gamma}_B^{(l)} = g_{\boldsymbol{\phi}}^{(l)}\left(\mathbf{h}_B^{(l)}\right). \tag{1}$$

An elementwise product is then performed between the scale vector and the original tensor

$$\hat{\mathbf{H}}_1^{(l)} = \mathbf{H}_1^{(l)} * \boldsymbol{\gamma}_1^{(l)} \quad \cdots \quad \hat{\mathbf{H}}_B^{(l)} = \mathbf{H}_B^{(l)} * \boldsymbol{\gamma}_B^{(l)}, \tag{2}$$

with the aim of modulating the activation along the channel dimension. This operation can be interpreted as a soft attention mechanism, with the MLP conditionally deciding which channel must be attended to. A graphical representation of SE is provided in Figure 2 (left).

**Contextual Squeeze-Excite (CaSE)** Standard SE is an instance-based mechanism that is suited for i.i.d. data in the supervised setting. In a meta-learning setting we can exploit the distinction in tasks to define a new version of SE for task-based channel-wise feature adaptation. For a task $\tau = (\mathcal{C}, \mathcal{T})$, consider the $N$ images from the context set $\mathcal{C}$, and the tensors produced by each convolution in the layers of interest $\{\mathbf{H}_1, \ldots, \mathbf{H}_N\}$ with $\mathbf{H}_i \in \mathbb{R}^{C \times H \times W}$. As in standard SE, we first apply a *spatial* pooling to each tensor $\mathbf{H}_i$ which produces $N$ vectors $\{\mathbf{h}_1, \ldots, \mathbf{h}_N\}$ of shape $\mathbf{h}_i \in \mathbb{R}^C$. Then a *context* pooling is performed; this corresponds to an empirical mean over $\{\mathbf{h}_1, \ldots, \mathbf{h}_N\}$ (see Appendix A for more details about context pooling). The pooled representation is passed to the associated MLP to produce a single scale-vector for that layer

$$\boldsymbol{\gamma}^{(l)} = g_{\boldsymbol{\phi}}^{(l)}\left(\bar{\mathbf{h}}^{(l)}\right) \quad \text{with} \quad \bar{\mathbf{h}}^{(l)} = \frac{1}{N}\left(\mathbf{h}_1^{(l)} + \cdots + \mathbf{h}_N^{(l)}\right), \tag{3}$$

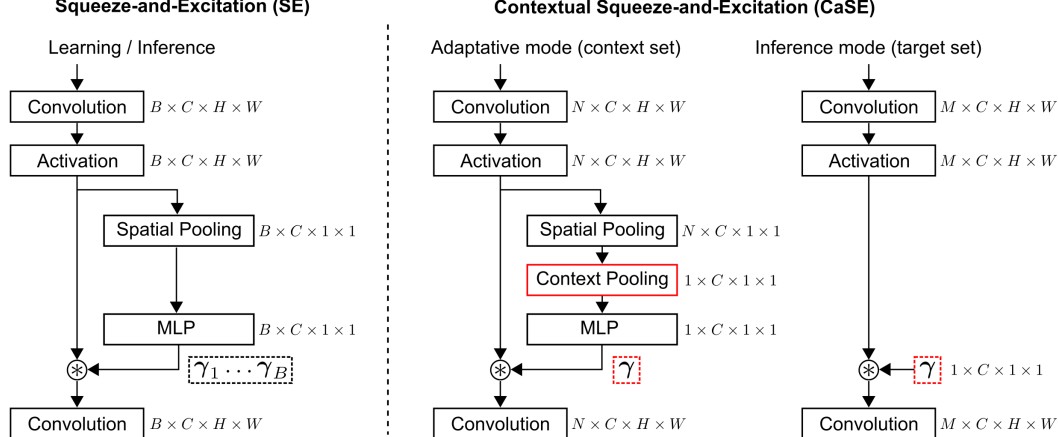

Figure 2: Comparison between the standard Squeeze-Excite (left) and the proposed Contextual Squeeze-Excite (right). Red frames highlight the two key differences between SE and CaSE: context pooling and scale transfer from context to target. $B$ = mini-batch size, $C$ = channels, $H$ = height, $W$ = width, $N$ = context-set size, $M$ = target-set size, $*$ elementwise multiplication.

which is then multiplied elementwise by the original tensors

$$\hat{\mathbf{H}}_1^{(l)} = \mathbf{H}_1^{(l)} * \boldsymbol{\gamma}^{(l)} \; \cdots \; \hat{\mathbf{H}}_N^{(l)} = \mathbf{H}_N^{(l)} * \boldsymbol{\gamma}^{(l)}. \tag{4}$$

The scale vector is estimated in adaptive mode and transferred to the target points $\mathcal{T}$ in inference mode (no forward pass on the MLPs), as shown in the rightmost part of Figure 2. In synthesis, the three major differences between SE and CaSE are: (i) CaSE uses a contextual pooling with the aim of generating an adaptive vector per-task instead of per-instance as in SE; (ii) CaSE distinguishes between an adaptive mode and an inference mode that transfers the scale from context to target, while SE does not make such a distinction; and (iii) CaSE parameters are estimated via episodic meta-training while SE parameters via standard supervised-training. In Section 5.1 we show that those differences are fundamental to achieve superior performance in the few-shot setting. A representation of a CaSE block is reported in Figure 2 (right), additional technical details are provided in Appendix A.

**Comparison with other adapters** Popular adaptation mechanisms for few-shot learning are based on Feature-wise Linear Modulation layers (FiLM, Perez et al. 2018). Those mechanisms perform adaptation using a separate convolutional set-encoder to produce an embedding of the context set. The embedding is forwarded to local MLPs to produce the scale and shift vectors of the FiLM layers that modulate a pretrained model. Variations of this adapter have been used in several methods, such as TADAM (Oreshkin et al., 2018), CNAPs (Requeima et al., 2019), SimpleCNAPs (Bateni et al., 2020), CAVIA (Zintgraf et al., 2019), and LITE (Bronskill et al., 2021). We will use the generic term *FiLM generator* to refer to these adapters and the term *FiLM* to refer to the scale and shift vectors used to modulate the activations. There are two key differences between FiLM and CaSE: (i) CaSE exploits context pooling to aggregate the activations of the backbone instead of a separate set-encoder as in FilM generators (see Appendix A for details) which is more efficient in terms of parameter count and implementation overhead; and (ii) FiLM uses scale and shift to modulate the activations, CaSE only the scale, therefore 50% less parameters are stored in memory and transferred during inference. In Section 5.1 we compare CaSE and the FiLM generators used in a recent SOTA method (LITE, Bronskill et al. 2021), showing that CaSE is superior in terms of accuracy while using a fraction of the amortization parameters.

## 3 UpperCaSE: system description and optimization protocol

We exploit CaSE blocks as part of UpperCaSE, a hybrid training protocol based on Coordinate-Descent (CD). We call this protocol *hybrid* because it combines a meta-training procedure to optimize the CaSE parameters (body) with a fine-tuning routine to estimate the task-specific parameters (head).

**Preliminaries** We are given a feature extractor (body) pretrained with supervised learning on a large dataset (e.g. ImageNet), defined as $b_{\boldsymbol{\theta}}(\cdot)$ where $\boldsymbol{\theta}$ are the pretrained parameters. CaSE blocks,

parameterized by $\phi$, are added to the model at specific locations to give $b_{\theta,\phi}(\cdot)$ (see Appendix A for details about this step). We are interested in learning the CaSE parameters $\phi$ keeping constant the pretrained parameters $\theta$ (omitted from here to keep the notation uncluttered). At training time, we are given a series of tasks $\tau = \{\mathcal{C}, \mathcal{T}\} \sim \mathcal{D}$, where $\mathcal{D}$ is the training set. The number of classes (way) is calculated from the context set and used to define a linear classification head $h_{\psi_\tau}(\cdot)$ parameterized by $\psi_\tau$. The complete model is obtained by nesting the two functions as $h_{\psi_\tau}(b_\phi(\cdot))$. We indicate a forward pass through the body over the context inputs with the shorthand $b_\phi(\mathcal{C}^x) \rightarrow \{\mathbf{z}_1, \ldots, \mathbf{z}_N\}$, where $\mathbf{z}_n$ is the context embedding for the input $\mathbf{x}_n$. All the context embeddings and the associated labels are stored in $\mathcal{M} = \{(\mathbf{z}_n, y_n)\}_{n=1}^N$.

**Optimization challenges** We have two sets of learnable parameters, $\phi$ the CaSE parameters, and $\psi_\tau$ the parameters of the linear head for the task $\tau$. While $\phi$ is shared across all tasks (task-common), $\psi_\tau$ must be inferred on the task at hand (task-specific). In both cases, the objective is the minimization of a classification loss $\mathcal{L}$. There are some challenges in optimizing the CaSE parameters in the body, as shown by the decomposition of the full gradient

$$\frac{d\mathcal{L}}{d\phi} = \sum_\tau \left( \frac{\partial \mathcal{L}_\tau}{\partial \psi_\tau} \frac{d\psi_\tau}{d\phi} + \frac{\partial \mathcal{L}_\tau}{\partial \phi} \right). \tag{5}$$

The first term $\partial \mathcal{L}_\tau / \partial \psi_\tau$ (sensitivity of the loss w.r.t. the head) and the direct gradient $\partial \mathcal{L}/\partial \phi$ (sensitivity of the loss w.r.t. the adaptation parameters with a fixed head) can be obtained with auto-differentiation as usual. The second term $d\psi_\tau/d\phi$ (sensitivity of the head w.r.t. the adaptation parameters) is problematic because $\psi_\tau$ is obtained iteratively after a sequence of gradient updates. Backpropagating the gradients to $\phi$ includes a backpropagation through all the gradient steps performed to obtain the task-specific $\psi_\tau$. Previous work has showed that this produces instability, vanishing gradients, and high memory consumption (Antoniou et al., 2018; Rajeswaran et al., 2019).

**Meta-training via Coordinate-Descent** A potential solution to these issues is the use of implicit gradients (Chen et al., 2020; Rajeswaran et al., 2019; Chen et al., 2022). The main problem with implicit gradients is the computation and inversion of the Hessian matrix as part of Cauchy's implicit function theorem, which is infeasible when the number of parameters in the linear head is large. Another possible solution is the use of an alternating-optimization scheme, similar to the one proposed in a number of recent methods such as MetaOptNet (Lee et al., 2019), R2D2 (Bertinetto et al., 2018), and ANIL (Raghu et al., 2019). These methods share the idea of inner-loop-head/outer-loop-body meta-training, and they find the parameters of the linear head with closed form solutions or by stochastic optimization. Starting from similar assumptions we propose a simple yet effective alternating-optimization scheme, which we formalize using Coordinate-Descent (CD) (Wright, 2015). The idea behind CD is to consider the minimization of a complex multi-variate function as a set of simpler objectives that can be solved one at a time. In our case, we can consider the joined landscape w.r.t. $\phi$ and $\psi_\tau$ as composed of two separate sets of coordinates (block CD, Wright 2015). By minimizing $\psi_\tau$ first, we reach a local minimum where $\partial \mathcal{L}_\tau / \partial \psi_\tau \approx 0$. Therefore CD induces a direct optimization objective w.r.t. $\phi$, with Equation (5) reducing to $\partial \mathcal{L}_\tau / \partial \phi$ (no red term). The time complexity of this method is only affected by the number of classes but is constant w.r.t. the number of training points due to the use of mini-batching, which scales well with large tasks (e.g. those in MetaDataset and VTAB). See Appendix B for more details.

In practice, at each training iteration we sample a task $\tau = (\mathcal{C}, \mathcal{T}) \sim \mathcal{D}$, perform a forward pass on the body (with CaSE in adaptive mode) to get

$$b_\phi(\mathcal{C}^x) \rightarrow \{\mathbf{z}_1, \ldots, \mathbf{z}_N\}. \tag{6}$$

The context embeddings are temporarily stored in a buffer with their associated labels $\mathcal{M} = \{(\mathbf{z}_n, y_n)\}_{n=1}^N$ to avoid expensive calls to $b_\phi(\cdot)$. We then set the head parameters to zero, and solve the first minimization problem (inner-loop), obtaining the task-specific parameters $\psi_\tau$ via

$$\psi_\tau \leftarrow G\left(\alpha, \mathcal{M}, \mathcal{L}, h_{\psi_\tau}\right) \tag{7}$$

where $\alpha$ is a learning rate, and $G(\cdot)$ is a functional representing an iterative gradient-descent routine for parameter estimation (e.g. maximum likelihood estimation or maximum a posteriori estimation). Note that the iterative routine in Equation (7) only relies on the head $h_{\psi_\tau}(\cdot)$ and not on the body $b_\phi(\cdot)$, which is the primary source of memory savings and the crucial difference with common fine-tuning methods. Moreover, the inner-loop is agnostic to the choice of optimizer, it can handle many gradient steps without complications, exploit parallelization and efficient mini-batching.

We then turn our attention to the second coordinate: the task-common parameters of the CaSE blocks in the body. For a single task, the update consists of a single optimization step w.r.t. $\phi$ (outer-loop) given support/target points and the task-specific parameters $\psi_\tau$ identified previously. The final form of the equation depends on the optimizer, for a generic SGD the update is given by

$$\phi \leftarrow \phi - \beta \nabla_\phi \mathcal{L} \left( \mathcal{C}^y \cup Q^y, h_{\psi_\tau}, b_\phi \right), \tag{8}$$

where $\beta$ is a learning rate. CaSE blocks must be in adaptive mode to allow the backpropagation of the gradients to the MLPs. The process repeats, alternating the minimization along the two sets of coordinates. The pseudo-code for train and test is provided in Appendix B.

**Inference on unseen tasks** After the training phase, we are given an unseen task $\tau_* = (\mathcal{C}_*, \mathbf{x}_*)$ where $\mathbf{x}_*$ is a single target input and $y_*$ the associate true label to estimate. Inference consists of three steps: (i) forward pass on the body for all the context inputs with CaSE set to adaptive mode as in Equation (6) and embeddings/labels stored in $\mathcal{M}$, (ii) estimation of the task-specific parameters $\psi_*$ via iterative updates as in Equation (7), and (iii) inference of the target-point membership via a forward pass over body and head $\hat{y}_* = h_{\psi_*} (b_\phi(\mathbf{x}_*))$ with CaSE in inference mode.

## 4   Related work

**Meta-learning** There has been a large volume of publications related to meta-learning. Here we focus on those methods that are the most related to our work, and refer the reader to a recent survey for additional details (Hospedales et al., 2020). LITE (Bronskill et al., 2021) is a protocol for training meta-learners on large images, that achieved SOTA accuracy on VTAB+MD. LITE is particularly relevant in this work, as its best performing method is based on Simple CNAPs (Bateni et al., 2020) that exploits FiLM for fast body adaptation. We compare against LITE in Section 5.2 showing that UpperCaSE is superior in terms of classification accuracy and parameter efficiency.

**Fine-tuning** Chen et al. (2019) were the first to expose the potential of simple fine-tuning baselines for transfer learning. MD-Transfer has been proposed in Triantafillou et al. (2019) as an effective fine-tuning baseline for the MetaDataset benchmark. More recently Kolesnikov et al. (2020) have presented Big Transfer (BiT), showing that large models pretrained on ILSVRC-2012 ImageNet and the full ImageNet-21k are very effective at transfer learning. MD-Transfer and BiT differ in terms of classification head, learning schedule, normalization layers, and batching. Fine-tuning only the last linear layer can be effective (Bauer et al., 2017; Tian et al., 2020). We compare against this baseline in Section 5.1, showing that adapting the body via CaSE significantly boosts the performance. Overall, fine-tuners have consistently outperformed meta-learners in terms of classification accuracy, only under particular conditions (e.g. strong class-imbalance) the trend is reversed (Ochal et al., 2021a,b).

**Hybrids** Hybrid methods are trained episodically like meta-learners but rely on fine-tuning routines for adaptation. Model Agnostic Meta-Learning (MAML, Finn et al. 2017) finds a set of parameters that is a good starting point for adaptation towards new tasks in a few gradient steps. MAML has been the inspiration for a series of other models such as MAML++ (Antoniou et al., 2018), ProtoMAML (Triantafillou et al., 2019), and Reptile (Nichol et al., 2018).

**Dynamic networks** CaSE blocks belong to the wider family of dynamic networks, models that can adapt their structure or parameters to different inputs (Han et al., 2021). Adaptive components have been used in a variety of applications, such as neural compression (Veit and Belongie, 2018; Wu et al., 2018), generation of artistic styles (Dumoulin et al., 2016; Huang and Belongie, 2017), or routing (Guo et al., 2019). Residual adapters (Rebuffi et al., 2017, 2018) have been used in transfer learning (non few-shot) but they rely on fine-tuning routines which are significantly slow during adaptation. More recently, Li et al. (2022) have used serial and residual adapters in the few-shot setting, with the task-specific weights being adapted from scratch on the context set. This approach has similar limitations, since it requires backpropagation to the task-specific weights in the body of the network which is costly. In Sun et al. (2019) the authors introduce a Meta-Transfer Learning (MTL) method for the few-shot setting. In MTL a series of scale and shift parameters are meta-learned across tasks and then dynamically adapted during the test phase via fine-tuning. This method suffers of similar limitations, as the fine-tuning stage is expensive during adaptation. Moreover, MTL relies on scale and shift vectors to perform adaptation whereas CaSE only relies on a scale vector, meaning that it needs to store and transfer 50% less parameters at test time.

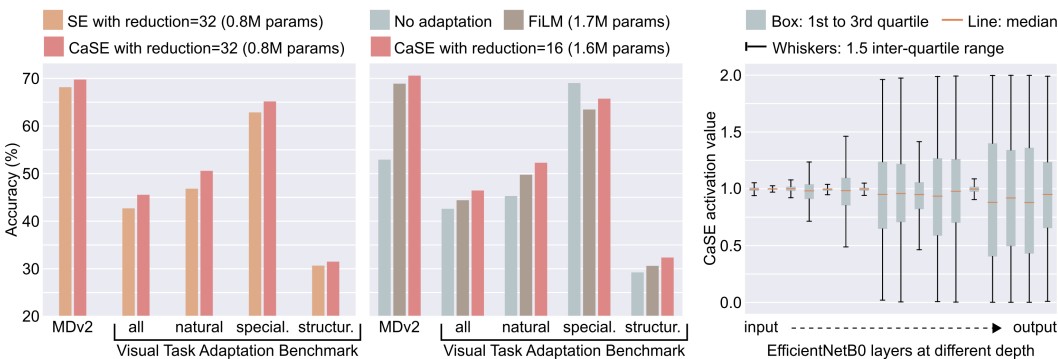

Figure 3: **Left**: CaSE vs Squeeze-and-Excitation (SE) (both methods use EfficientNetB0, $84 \times 84$ inputs, Mahalanobis-distance head). CaSE outperforms SE in all conditions. **Center**: CaSE vs. FiLM generators (Bronskill et al., 2021) and a baseline with no body adaptation (all methods use EfficientNetB0, $84 \times 84$ inputs, Mahalanobis-distance head). CaSE outperforms FiLM generators in all conditions. **Right**: boxplot of CaSE activations at different depth of an EfficientNetB0 for 800 tasks sampled from the MDv2 test set ($224 \times 224$ inputs, UpperCaSE). The modulation of CaSE is minimal at early stages for general-purpose filters and increases at deeper stages.

# 5 Experiments

In this section we report on experiments on VTAB+MD (Dumoulin et al., 2021) and ORBIT (Massiceti et al., 2021). VTAB+MD has become the standard evaluation protocol for few-shot approaches, and it includes a large number of datasets (8 test dataset for MD, 18 for VTAB). For a description of ORBIT, see Section 5.2. In all experiments we used the following pretrained (on ImageNet) backbones: EfficientNetB0 from the official Torchvision repository; ResNet50x1-S released with BiT (Kolesnikov et al., 2020). We used three workstations (CPU 6 cores, 110GB of RAM, and a Tesla V100 GPU), the meta-training protocol of Bronskill et al. (2021) ($10K$ training tasks, updates every 16 tasks), the Adam optimizer with a linearly-decayed learning rate in $[10^{-3}, 10^{-5}]$ for both the CaSE and linear-head. The head is updated 500 times using a random mini-batch of size 128. MD test results are averaged over 1200 tasks per-dataset (confidence intervals in appendix). We did not use data augmentation. Code to reproduce the experiments is available at `https://github.com/mpatacchiola/contextual-squeeze-and-excitation`.

## 5.1 Analysis of CaSE blocks

In this sub-section we report empirical results related to CaSE blocks in three directions: **1)** we compared standard SE (Hu et al., 2018) and CaSE on MDv2 and VTAB, confirming that a) adaptation helps over not adapting, b) contextual adaptation (CaSE) outperforms instance based adaptation (SE); **2)** we compare CaSE against a SOTA FiLM generator (Bronskill et al., 2021), showing that CaSE is significantly more efficient using 75% fewer parameters while boosting the classification accuracy on average by $+1.5\%$ on VTAB and MD2; and **3)** we provide an insight on the effectiveness of CaSE blocks with a series of qualitative analysis.

**Comparing SE vs. CaSE** We compare standard SE and the proposed CaSE on VTAB and MD-v2. For a fair comparison we keep constant all factors of variation (backbone, training schedule, hyperparameters, etc.) and use the same reduction of 32 (0.8M adaptive parameters). In order to compare the results with the other experiments in this section, we use a Mahalanobis-distance head as in Bronskill et al. (2021), reporting results with a linear head in the appendix. We summarize the results in Figure 3 (left) and add a tabular breakdown in Appendix C.2. CaSE outperforms SE in all conditions, confirming that a contextual adaptation mechanism is fundamental to transfer knowledge effectively across tasks.

**Comparing adaptation mechanisms** We perform a comparison on VTAB+MD of CaSE against FiLM generators (Bronskill et al., 2021), and a baseline that uses a pretrained model but no adaptation of the body. Methods are compared in identical conditions, using a Mahalanobis-distance head, an EfficientNetB0 backbone, and same training schedule. We show a summary of the results in

Figure 3 (center) and provide a full breakdown in the appendix. CaSE is able to outperform FiLM generators in all conditions. In Appendix C.3 we report the results for CaSE with reduction 64 (0.4M parameters) showing that it is able to outperform FiLM generators (1.7M parameters) using a fraction of the parameters. The comparison with the baseline with no adaptation, shows that in all but one condition (VTAB specialized) adaptation is beneficial. This is likely due to the strong domain shift introduced by some of the specialized datasets.

**Role of CaSE blocks** To examine the role of CaSE blocks we analyze the aggregated activations at different stages of the body for 800 tasks sampled from the MDv2 test set using an EfficientNetB0 trained with UpperCaSE on $224 \times 224$ images. In Figure 3 (right) we report the aggregated distribution as boxplots, and in Appendix C.5 we provide a per-dataset breakdown. Overall the median is close to 1.0 (identity) which is the expected behavior as on average we aim at exploiting the underlying pretrained model. The variance is small at early stages, indicating that CaSE has learned to take advantage of general-purpose filters that are useful across all tasks. In deeper layers the variance increases, showing a task-specific modulation effect. In Appendix C.5 we also include a plot with per-channel activations for all datasets at different depths, showing that the modulation is similar across datasets at early stages and it diverges later on. An ablation study of different factors (e.g. reduction, number of hidden layers, activation functions) is reported in Appendix C.4.

## 5.2 Performance evaluation of UpperCaSE

In this sub-section we analyze the performance of UpperCaSE in two settings: **1)** comparison on the VTAB+MD benchmark against SOTA fine-tuners and meta-learners, where we show that UpperCaSE is able to outperform all the meta-learners, narrowing the gap with Big Transfer (BiT) on VTAB; **2)** we show an application of UpperCaSE in a real-world personalization task on the challenging ORBIT dataset (Massiceti et al., 2021) for the cross-domain condition MDv2→ORBIT, where we achieve the best average-score in most metrics, although these improvements are within the error bars.

Table 1: **UpperCaSE outperforms fine-tuners on MDv2 and narrows the gap on VTAB with the leading method (BiT) with a much lower adaptation cost**. Average accuracy on the 26 datasets of VTAB+MD. RN=ResNet, EN=EfficientNet. Img: image size. Param.: total parameters (no adapters) in millions. Cost: MACs to adapt on a task (10-shot, 100-way), in Teras. Best results in bold.

| Method | Protocol | Net | Img | Param. | Cost ↓ MACs | MDv2 ↑ all | VTAB ↑ all | natur. | spec. | struc. |
|---|---|---|---|---|---|---|---|---|---|---|
| MD-Transfer | fine-tuning | RN18 | 126 | 11.2 | 118.6 | 63.4 | 55.6 | 52.4 | 72.9 | 49.3 |
| SUR | fine-tuning | RN50 | 224 | 164.6 | 28.8 | 71.3 | 43.7 | 50.9 | 66.2 | 27.2 |
| Big Transfer | fine-tuning | RN50 | 224 | 23.5 | 526.3 | 73.3 | **65.4** | **69.4** | **81.0** | **54.5** |
| *UpperCaSE* | hybrid | RN50 | 224 | 23.5 | 0.8 | 74.9 | 56.6 | 66.3 | 80.1 | 37.6 |
| *UpperCaSE* | hybrid | ENB0 | 224 | 4.0 | **0.2** | **76.1** | 58.4 | 69.1 | 80.3 | 39.4 |

Table 2: **UpperCaSE outperforms all meta-learning/hybrid methods and uses the lowest number of parameters per adaptive blocks**. Average accuracy on the 26 datasets of VTAB+MD. RN=ResNet, EN=EfficientNet. Img: image size. Param.: total parameters (excluding adapters). Adapt.: total adaptive parameters in millions. Best results in bold.

| Method | Protocol | Net | Img | Param. | Adapt. ↓ count | MDv2 ↑ all | VTAB ↑ all | natur. | spec. | struc. |
|---|---|---|---|---|---|---|---|---|---|---|
| ProtoMAML | hybrid | RN18 | 126 | 11.2 | n/a | 64.2 | 45.0 | 45.7 | 70.7 | 31.5 |
| CTX | meta-learning | RN34 | 224 | 21.3 | n/a | 71.6 | 50.5 | 61.1 | 67.3 | 34.0 |
| ProtoNet | meta-learning | ENB0 | 224 | 4.0 | n/a | 72.7 | 46.1 | 60.9 | 64.2 | 25.9 |
| LITE | meta-learning | ENB0 | 224 | 4.0 | 1.7 | 73.8 | 51.4 | 65.2 | 71.9 | 30.8 |
| *UpperCaSE* | hybrid | RN50 | 224 | 23.5 | 0.8 | 74.9 | 56.6 | 66.3 | 80.1 | 37.6 |
| *UpperCaSE* | hybrid | ENB0 | 224 | 4.0 | **0.4** | **76.1** | **58.4** | **69.1** | **80.3** | **39.4** |

**Comparison on VTAB+MD** We compare UpperCaSE against fine-tuners, meta-learners, and hybrids on the 18 datasets of VTAB and the 8 datasets of MetaDataset-v2 (MDv2) and report the results

Table 3: **ORBIT: UpperCaSE obtains the best average-score in most metrics, being within error bars with leading methods.** Average accuracy and 95% confidence interval for frames, videos, and frames to recognition (FTR). Cost: average MACs over all tasks (Teras). Results and setup from Massiceti et al. (2021): meta-train on MetaDataset and test on ORBIT, image-size $84 \times 84$, ResNet18 backbone, 85 test tasks (17 test users, 5 tasks per user). Best results (within error bars) in bold.

| | Cost | Clean Video Evaluation (CLE-VE) | | | Clutter Video Evaluation (CLU-VE) | | |
|---|---|---|---|---|---|---|---|
| Method | MACs↓ | frame acc.↑ | FTR↓ | video acc.↑ | frame acc.↑ | FTR↓ | video acc.↑ |
| ProtoNet | **3.2** | **59.0±2.2** | **11.5±1.8** | 69.2±3.0 | 47.0±1.8 | 20.4±1.7 | 52.8±2.5 |
| CNAPs | 3.5 | 51.9±2.5 | 20.8±2.3 | 60.8±3.2 | 41.6±1.9 | 30.7±2.1 | 43.0±2.5 |
| MAML | 95.3 | 42.5±2.7 | 37.3±3.0 | 47.0±3.2 | 24.3±1.8 | 62.3±2.3 | 25.7±2.2 |
| FineTuner | 317.7 | **61.0±2.2** | **11.5±1.8** | **72.6±2.9** | **48.4±1.9** | **19.1±1.7** | **54.1±2.5** |
| *UpperCaSE* | 3.5 | **63.0±2.2** | **8.8±1.6** | **74.4±2.8** | **48.1±1.8** | **18.2±1.7** | **54.5±2.5** |

in Table 1 and Table 2. UpperCaSE outperforms all methods (including BiT) on MDv2 with an accuracy of 74.9% (ResNet50) and 76.1% (EfficientNetB0). On VTAB, UpperCaSE outperforms most methods, narrowing the gap with BiT. A closer look at the differences in performance on VTAB between UpperCaSE and BiT (see Table 1) shows that the gap is narrower on the natural and specialized splits ($+3.1\%$ and $+0.9\%$) but larger on structured ($+16.9\%$).

The breakdown by dataset reported in Appendix C.6 shows that the major performance drops are on tasks that require localization and counting (e.g. dSprites, SmallNORB). Similar issues are encountered by methods such as LITE (Bronskill et al., 2021) which are based on FiLM generators, suggesting that those tasks may introduce a strong domain shift w.r.t. the meta-training set that is difficult to compensate without fine-tuning the body. It is not clear whether transfer learning is beneficial on these datasets in the first place. The results in terms of adaptation cost (see Table 1) over a synthetic task (10-shot, 100 way) show that UpperCaSE is orders of magnitude more efficient ($0.2 \times 10^{12}$ MACs) than all fine-tuners, with BiT being the most expensive method overall ($526.3 \times 10^{12}$ MACs). The comparison against meta-learners in terms of number of adaptive parameters (see Table 2) shows that UpperCaSE requires a fraction of the parameters (0.4 vs 1.7 millions for an EfficientNetB0) compared to LITE (Bronskill et al., 2021) which is based on FiLM generators.

**Comparison on ORBIT** We compare UpperCaSE to other methods on ORBIT (Massiceti et al., 2021), a real-world dataset for teachable object recognizers. ORBIT consists of 3822 videos of 486 objects recorded by 77 blind/low-vision people on their mobile phones. The dataset is challenging because objects are poorly framed, occluded, blurred, and in a wide variation of backgrounds and lighting. The dataset includes two sets of target videos, one for clean video evaluation (CLE-VE) with well-centered objects, and another for clutter video evaluation (CLU-VE) with objects in complex, cluttered environments. We consider a hard transfer-learning condition where classifiers are meta-trained on MetaDataset and tested on ORBIT.

Results are reported in Table 3. UpperCaSE outperforms all other methods (on average) on most metrics, being within error bars with the two leading methods. Comparing UpperCaSE with FineTuner, the gap in favor of UpperCaSE is marginal on CLU-VE but substantial on CLE-VE (frame accuracy $+2\%$, video accuracy $+1.8\%$, and FTR $-2.7$). Comparison in terms of adaptation cost (average MACs over all tasks) shows that UpperCaSE is orders of magnitude more efficient than FineTuner and close to the leading method (ProtoNet).

## 6 Conclusions

We have introduced a new adaptive block called CaSE, which is based on the popular Squeeze-and-Excitation (SE) block proposed by Hu et al. (2018). CaSE is effective at modulating a pretrained model in the few-shot setting, outperforming other adaptation mechanisms. Exploiting CaSE we have designed UpperCaSE, a hybrid method based on a Coordinate-Descent training protocol, that combines the performance of fine-tuners with the low adaptation cost of meta-learners. UpperCaSE achieves SOTA accuracy w.r.t. meta-learners on the 26 datasets of VTAB+MD and it compares favorably with leading methods in the ORBIT personalization benchmark.

**Limitations** There are two *limitations* that are worth mentioning: (i) UpperCaSE requires iterative gradient updates that are hardware-dependent and may be slow/unavailable in some portable devices; (ii) breakdown VTAB results per-dataset shows that the method falls short on structured datasets. This indicates that fine-tuning the body may be necessary for high accuracy when the shift w.r.t. the meta-training set is large.

**Societal impact** Applications based on CaSE and UpperCaSE could be deployed in few-shot classification settings that can have a positive impact such as: medical diagnosis, recommendation systems, object detection, etc. The efficiency of our method can reduce energy consumption and benefit the environment. Certain applications require careful consideration to avoid biases that can harm specific groups of people (e.g. surveillance, legal decision-making).

## Acknowledgments and Disclosure of Funding

Funding in direct support of this work: Massimiliano Patacchiola, John Bronskill, Aliaksandra Shysheya, and Richard E. Turner are supported by an EPSRC Prosperity Partnership EP/T005386/1 between the EPSRC, Microsoft Research and the University of Cambridge. The authors would like to thank: anonymous reviewers for useful comments and suggestions; Aristeidis Panos, Daniela Massiceti, and Shoaib Ahmed Siddiqui for providing suggestions and feedback on the preliminary version of the manuscript.

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
