# OpenReview forum: "Contextual Squeeze-and-Excitation for Efficient Few-Shot Image Classification"
_NeurIPS.cc/2022/Conference — NeurIPS 2022 Accept_

### Official Review · Reviewer_FFac · 2022-06-30

**Rating:** 5
**Confidence:** 4
**Soundness:** 3 good
**Presentation:** 3 good
**Contribution:** 1 poor

**Summary:**

This paper mainly focuses on few-shot learning. The authors advocate the importance of an efficient finetuning algorithm for FSL. To the end, they propose a new module based on SE block, which generates scaling parameters using task-specific information. Furthermore, they propose to leverage coordinate-descent in meta-training to solve the problems of instability, vanishing gradients and high memory consumption.

**Questions:**

1. The main concern comes from the proposed CaSE. This module takes episode-level features as input and provides channel-wise scaling to the final features, which reminds me of MTL [1]. The MTL is also aimed to meta-train new modules, that produce scaling and shifting parameters, and use them directly in testing phase. I find it very similar to the proposed method. It would be better if the authors could explain the exact difference between two methods and the significance.
2. Beyond MTL, there are other works that can fulfill the goal of finetuning only few parameters on each episode, such as [2]. The authors should provide more discussion and comparison.
3. I wonder if the authors can provide results on smaller dataset like miniImageNet so that the method can be compared with more state-of-the-art methods.


[1] Sun Q, Liu Y, Chua T S, et al. Meta-transfer learning for few-shot learning[C]//Proceedings of the IEEE/CVF Conference on Computer Vision and Pattern Recognition. 2019: 403-412.

[2] Li W H, Liu X, Bilen H. Cross-domain Few-shot Learning with Task-specific Adapters[C]//Proceedings of the IEEE/CVF Conference on Computer Vision and Pattern Recognition. 2022: 7161-7170.

**Limitations:**

Detailed comments are in the questions. In summary, I find the proposed module similar to the existing method in FSL, which is the main problem of the method.

**Strengths And Weaknesses:**

1. The authors conduct extensive experiments to show the effectiveness of their method.
2. The proposed coordinate-descent for meta-learning is interesting and can be a potential plug-in solver for other meta-learning methods.

---

> ### Author Response · Authors · 2022-07-27
> **Missing references**
>
> We thank the reviewer for the constructive feedback.
>
> We have noticed that the reviewer is mentioning two papers using the numbered references [1] and [2] (in the "Questions" section), but those references are missing. We kindly ask if those references can be provided, since they are key elements in the questions and they are needed for the rebuttal.

---

> ### Author Response · Authors · 2022-08-01
> **Answer to reviewer FFac**
>
> We thank the reviewer for the useful feedback and for reviewing our paper.
>
> **1. “This module takes episode-level features as input and provides channel-wise scaling to the final features, which reminds me of MTL [1]. The MTL is also aimed to meta-train new modules, that produce scaling and shifting parameters, and use them directly in testing phase. I find it very similar to the proposed method. It would be better if the authors could explain the exact difference between two methods and the significance.”**
>
> Thank you for pointing out this relevant paper. We have read the paper and identified some important differences that are here detailed.
>
> The first difference between MTL and CaSE  is that MTL meta-learns the scale and shift and then uses them during the test phase without any dynamic adaptation. On the contrary, our method uses compact fully-connected networks to dynamically generate the scale vector based on the context set. We believe that dynamic adapters are a crucial ingredient in meta-learners. This is supported by the empirical results we have showed in Section 5.1, where our method outperforms non-dynamic baselines (e.g. see Figure 3, no-adaptation baseline). In addition, we note that other methods based on dynamic FiLM adapters (e.g. TADAM, SimpleCNAPs, LITE) have shown state-of-the-art accuracy on a variety of benchmarks, further supporting our claim.
>
> The second difference between MTL and CaSE is that MTL relies on a shift and scale vector while CaSE only relies on a scale vector. This means that CaSE stores in memory and transfer 50% less parameters than MTL during the inference step.
>
> We will integrate these comparisons to MTL in the paper.
>
> **2. “Beyond MTL, there are other works that can fulfill the goal of finetuning only few parameters on each episode, such as [2]. The authors should provide more discussion and comparison.”**
>
> We thank the reviewer for pointing out this interesting paper. The method proposed in [2] can be considered as a fine-tuning method since it learns task-specific weights from scratch without dynamically estimating them. Therefore, this approach is substantially different from our proposal since we dynamically estimate the task-specific weights (scale vector) in a single forward pass on the context set without the need of expensive fine-tuning routines over the network body. Our approach has a far lower computational complexity to adapt to a new task. We consider this aspect crucial because our contribution is mainly oriented towards improving efficiency in terms of adaptation time without sacrificing accuracy.
>
> The authors of [2] reported results on MetaDataset with a ResNet34 obtaining an average score of 73.9 on the 8 datasets of MDv2 (with large images of size 224x224). Our method performs better with an accuracy of 76.1 (with EfficientNetB0) and 74.9 (with ResNet50). We stress again the fact that our method is significantly faster during inference, since it does not need to fine-tune any parameter in the body of the network (those are generated in one step by the CaSE adapters).
>
> We will discuss [2] and report those considerations in the revised version of the paper.
>
>
>
> **3. “I wonder if the authors can provide results on smaller dataset like miniImageNet so that the method can be compared with more state-of-the-art methods.”**
>
> Our method exploits backbones that have been pretrained on ImageNet. Using those backbones on miniImageNet tasks will likely bias the results because there would be classes shared between training and test sets. This would give our method an unfair advantage and therefore we think is not appropriate.
>
> We also point out that the most recent state-of-the-art methods have been mainly tested on MetaDataset and VTAB which are more comprehensive benchmarks than miniImageNet. In the experimental section we have directly compared against these recent state-of-the-art methods (e.g. Big Transfer and LITE). Moreover, there is substantial evidence that shows how miniImageNet can be problematic for the evaluation of few-shot learning methods, this has been pointed out and empirically analyzed in a recent paper [3].
>
>
> **References**
>
> [1] Sun Q, Liu Y, Chua T S, et al. “Meta-transfer learning for few-shot learning”, Proceedings of the IEEE/CVF Conference on Computer Vision and Pattern Recognition. 2019: 403-412.
>
> [2] Li W H, Liu X, Bilen H. “Cross-domain Few-shot Learning with Task-specific Adapters”, Proceedings of the IEEE/CVF Conference on Computer Vision and Pattern Recognition. 2022: 7161-7170.
>
> [3] Huang, G., Larochelle, H., & Lacoste-Julien, S. (2019). “Are few-shot learning benchmarks too simple? solving them without task supervision at test-time". arXiv preprint arXiv:1902.08605.

---

> > ### Comment · Reviewer_FFac · 2022-08-07
> > **Thank you for your response**
> >
> > Thank the authors for the detailed response. I have several further comments.
> >
> > 1. About MTL
> >
> > The authors take MTL as a non-dynamic method, which is somehow inappropriate.  As far as I can understand, the scaling and shifting parameters in MTL are finetuned based on the support set of each episode, which means these parameters can well adapt to the support knowledge, playing the same role as the CaSE module.
> >
> > 2. Comparison with [2]
> >
> > The authors try to show the advantage against [2] by comparing the results on MD. However the setting is not the same. In MD+VTAB the whole ImageNet training set is used for training, while in MD only a meta-train split set is used for training to avoid category overlap.
> >
> > 3. Necessity of ImageNet pretrain
> >
> > In fact it is not well explained why the ImageNet pretrained model is necessary. There is no evidence that the CaSE module relies on the ImageNet training set and can easily fail given insufficient base data. The proposed module is simple enough to be directly plugged into any backbones. In the most straightforward case, replacing the TADAM module with CaSE can lead to experiment results on smaller datasets like miniImageNet.

---

> > > ### Author Response · Authors · 2022-08-07
> > > **Answer to Reviewer FFac (part 1)**
> > >
> > > Thank you for engaging in the discussion, we are happy to provide additional information and clarifications.
> > >
> > > **1. “About MTL. The authors take MTL as a non-dynamic method, which is somehow inappropriate. As far as I can understand, the scaling and shifting parameters in MTL are finetuned based on the support set of each episode, which means these parameters can well adapt to the support knowledge, playing the same role as the CaSE module.”**
> > >
> > > Thank you for pointing out this inconsistency. We have found the MTL paper to be unclear regarding this point. Our understanding was that MTL meta-learns the scale and shift but does not adapt them during the test phase.
> > >
> > > If the scale and shift parameters are finetuned based on the support set, as the reviewer suggests, then MTL is very similar to many other fine-tuning methods such as [2][4][5] and aligned with the class of fine-tuning methods we have compared against in the paper (e.g. Big Transfer, MD-Transfer, SUR). We have showed empirically in the paper (Section 5.2) that our method is significantly better than this class of methods in terms of accuracy/adaptation-cost.
> > >
> > > If MTL is part of the fine-tuning methods, then its main weakness would be the need for backpropagation to the body of the network (where scale and shift are located), that is required for fine-tuning the task-specific parameters. Our approach is much more efficient from this point of view, since the body of the network is not adapted by multiple fine-tuning steps but by a single forward step (with CaSE adaptation). In other words, by using CaSE we can generate the task-speficic parameters in one forward step over the body, while MTL requires multiple backward/forward steps on the entire backbone.
> > >
> > > Note that, independently from the dynamic or non-dynamic nature of MTL, the second difference w.r.t. CaSE remains. MTL uses a scale and shift vector for adaptation while CaSE only a scale vector, meaning that CaSE needs 50% less parameters to store in memory and transfer during inference, which further support our claim of superior efficiency.
> > >
> > > We are going to amend the paragraph related to MTL in the paper adding these considerations.
> > >
> > > **2. “Comparison with [2]. The authors try to show the advantage against [2] by comparing the results on MD. However the setting is not the same. In MD+VTAB the whole ImageNet training set is used for training, while in MD only a meta-train split set is used for training to avoid category overlap.”**
> > >
> > > We agree with the reviewer that this is not an apples-to-apples comparison for various reasons (training regime, hyper-parameters, backbone type, ImageNet split, etc) and we should have been clearer in motivating our answer in the rebuttal. In practice, it is hard to directly compare against [2] since the authors only released models that have been trained following the standard MD setup (not MD+VTAB).  In the rebuttal, we have reported the figures related to the setting that is more similar to the one we have used (in terms of backbone and test datasets), with the aim of showing that the accuracies are similar.
> > >
> > > We would like to stress again the fact that our focus on the paper is not only the accuracy but the tradeoff of accuracy/adaptation-cost. With respect to this tradeoff [2] is clearly not optimal, since it requires fine-tuning the task-specific parameters (located in the body of the network) via multiple forward/backward steps. This means that [2] is more expensive in terms of adaptation cost.
> > >
> > > To make this clear, we provide a comparison of [2] and our method in terms of Multiple-Accumulate operations (MACs). Note that the MACs count is independent from the training regime and base dataset since it only depends on the input/task size and backbone. Therefore, it is much easier to compare against this factor. We used a ResNet34 backbone for both methods, and the same settings we have described in our paper in Appendix C.1 regarding image/task size. The task-specific parameters in [2] are updated for 40 steps (as mentioned by [2] in Appendix C.5). The results show that [2] is two orders of magnitude more expensive than our method, with a total number of MACs equal to 37.6T for [2] and equal to 0.5T for our method.
> > >
> > > We hope that this additional comparison will make clearer that our method is significantly more performant than [2] on the Pareto frontier of accuracy vs. adaptation cost.

---

> > > > ### Author Response · Authors · 2022-08-07
> > > > **Answer to Reviewer FFac (part 2)**
> > > >
> > > > **3. “In fact it is not well explained why the ImageNet pretrained model is necessary. […]”**
> > > >
> > > > Thank you for pointing this out, we agree with the reviewer that this needs clarifications.
> > > >
> > > > The use of a pretrained model for the few-shot setting is a practice that has been extensively used in recent state-of-the-art methods such as Big Transfer (BiT), CNAPs, SimpleCNAPs, and LITE. As far as we know, this practice has not been explicitly analyzed by the community but its benefits in terms of performance are empirically clear. We think that the use of a pre-trained model brings two advantages:
> > > >
> > > > 1. Pretrained models are better from a practical point of view. For instance, (i) it is possible to exploit large open-source models without the need of training from scratch, and (ii) they can be trained more easily using a supervised routine, parallelization on multi-GPUs, and standard datasets.
> > > > 2. Improved training stability. The joint optimization of the backbone and the adaptation mechanism can produce instability. In our experience this is often the case when training systems based on FiLM-generator adapters (e.g. CNAPs, SimpleCNAPs, LITE). We have noticed that in these models using a pretrained model solves this issue making the training stable.
> > > >
> > > > We believe that those two factors play an important role in the top-performances showed by methods that rely on pretrained models. Based on these considerations we have used a similar approach in CaSE. We will add those considerations in the paper.
> > > >
> > > > **4. “[…] There is no evidence that the CaSE module relies on the ImageNet training set and can easily fail given insufficient base data. […]”**
> > > >
> > > > We are not sure to have understood this part of the comment, we apology in advance if there is any misunderstanding. CaSE is forced to rely on the ImageNet training set by construction. CaSE is a modulator, meaning that it cannot solve a classification task on its own, but it must rely on an underlying pretrained model. The pretrained model has been trained on ImageNet. Therefore, CaSE relies on ImageNet.
> > > >
> > > > To be more precise, this is enforced by the training pipeline. The meta-training phase used to train the CaSE generator does not modify the weights of the pretrained model (trained on ImageNet), it means that the only knowledge that CaSE can exploit is the knowledge implicit in the backbone connections.
> > > >
> > > > Note that the CaSE modulation effect is fundamental and the pretrained model on its own is not sufficient. This has been empirically demonstrated in Section 5.1 and Figure 3 (center plot) where we show that the use of the same pretrained backbone leads to different results when no adaptation is used and when CaSE is added to the model. In the plot this is represented by the grey bar (no-adaptation) vs. the red bar (CaSE). As it is possible to see, the modulation effect of CaSE provides a clear advantage in most conditions.
> > > >
> > > > **5. “[…] The proposed module is simple enough to be directly plugged into any backbones. In the most straightforward case, replacing the TADAM module with CaSE can lead to experiment results on smaller datasets like miniImageNet.”**
> > > >
> > > > We agree with the reviewer that CaSE is flexible enough to be used in any backbone and that it could be a possible alternative to the FiLM-generator used in TADAM for datasets like miniImageNet. We think that the usefulness of an experiment like this could be limited. As we explained in the rebuttal, the use of miniImageNet has been seriously put into question (see [3] for a discussion) and we think that as a community we should move towards more challenging datasets such as the one we have used (MD and VTAB). The most recent state-of-the-art methods (BiT and LITE) rely on MD+VTAB as benchmark and discard miniImageNet. We have directly compared against BiT and LITE in the paper, showing convincingly that our method is comparable/better in terms of accuracy.
> > > >
> > > > **References**
> > > >
> > > > [1] Sun Q, Liu Y, Chua T S, et al. “Meta-transfer learning for few-shot learning”, Proceedings of the IEEE Conference on Computer Vision and Pattern Recognition. 2019.
> > > >
> > > > [2] Li W H, Liu X, Bilen H. “Cross-domain Few-shot Learning with Task-specific Adapters”, Proceedings of the IEEE Conference on Computer Vision and Pattern Recognition. 2022.
> > > >
> > > > [3] Huang, G., Larochelle, H., & Lacoste-Julien, S. (2019). “Are few-shot learning benchmarks too simple? solving them without task supervision at test-time".
> > > >
> > > > [4] Rebuffi, S.-A., Bilen, H., and Vedaldi, A. (2017). Learning multiple visual domains with residual adapters. In Advances in Neural Information Processing Systems.
> > > >
> > > > [5] Rebuffi, S.-A., Bilen, H., and Vedaldi, A. (2018). Efficient parametrization of multi-domain deep neural networks. In Proceedings of the IEEE Conference on Computer Vision and Pattern Recognition.
> > > >
> > > > [6] Bronskill, J., Massiceti, D., Patacchiola, M., Hofmann, K., Nowozin, S., and Turner, R. (2021). Memory efficient meta-learning with large images. Advances in Neural Information Processing Systems.

---

> > > > > ### Comment · Reviewer_FFac · 2022-08-09
> > > > > **Thank you for your response**
> > > > >
> > > > > Actually the claim is not so convincing that MD+VTAB is more valuable than miniImageNet and tieredImageNet in terms of FSL benchmark. In my opinion these datasets testify different aspects of a method. Sometimes those methods performing well on large-scale base data can easily fail on limited base data. However, given that most of my questions are solved, I will raise my score.

---

> > > > > > ### Author Response · Authors · 2022-08-09
> > > > > > **Thank you**
> > > > > >
> > > > > > We thank the reviewer for engaging in the rebuttal session and for the useful feedback. We are glad to know that most of the questions have been solved. We have different views on some aspects but overall the confrontation has improved the quality and clarity of the paper.

---

### Official Review · Reviewer_9nir · 2022-07-11

**Rating:** 7
**Confidence:** 4
**Soundness:** 4 excellent
**Presentation:** 4 excellent
**Contribution:** 3 good

**Summary:**

The work proposes a novel method for few-shot image classification. The method is named UpperCaSE and is based on adaptation of the Squeeze-and-Excitation block to learn task contextual parameters. Their approach is hybrid in terms of a combination between meta-learning and fine-tuning of the network. The proposed hybrid approach aims to bridge the gap between fine-tuning approaches which are more accurate, and the meta-learning approaches which have lower adaptation cost. Optimization protocol is based on Coordinate-Descent between the CaSE blocks in the network body (cross task parameters) and the task specific parameters of the head (last linear layer). The approach achieves new sota results for meta-learners on the VTAB+MD and ORBIT.

**Questions:**

I would have like to know how much the context pooling layer in the CaSE block is sensitive to the design choices. For example, have you tried to use several layers? adding a BN layer or other thing that either did not change the outcome or did not work at all.

**Limitations:**

I believe that the authors adequately addressed the limitations and potential negative societal impact of their work.
I appreciate for being honest about the lower results on structure datasets and the hypothesis that this case might require fine-tuning of the network's body.

**Strengths And Weaknesses:**

**Strengths**
- The paper is written in a clear and easy to follow manner.
- The method requires only a single forward pass over the context.
- The methods is simple and novel yet achieves very good results compared to other meta-learners.
- Experimentation study is comprehensive and also shows the downside on structured tasks.

**Weaknesses**
- I would have liked to see some ablation study and discussion on the CaSE block architecture choice.

---

> ### Author Response · Authors · 2022-08-01
> **Answer to reviewer 9nir**
>
> Thank you for reviewing our paper and for the positive feedback.
>
> **1. “I would have like to know how much the context pooling layer in the CaSE block is sensitive to the design choices. For example, have you tried to use several layers? adding a BN layer or other thing that either did not change the outcome or did not work at all.”**
>
> In a preliminary phase we have tried different configurations of CaSE blocks. In the paper, we have reported only the factors that played a major role. We will include the other experiments in the revised version. Here is an overview of what we have tried:
>
> - Activation function for the output layer of a CaSE block. This is the most important factor for a high accuracy. It is important for the initial output to be centred at 1 to enforce the identity function. To achieve this, we have used a sigmoid and multiplied its output with a constant value set to 2. The empirical results about this ablation are in Appendix A.1 (see Table 4).
>
> - Activation function for the hidden layers. CaSE seems quite robust against this factor. We have tried to use Tanh, ReLU, Leaky ReLU, and SiLU and all of them produced similar results. We have chosen SiLU for the experiments as this is the same activation typically used in Squeeze-and-Excitation layers (e.g. in EfficientNet backbones).
>
> - Number of hidden units in the hidden layers. This value is controlled by the “reduction” and “min_units” parameters in the code and it depends on the number of inputs. For example, if the inputs are 512 and we use a reduction of 32 the number of hidden units will be 512/32=16, this figure will be set to be equal to min_units if 16 is less than min_units. We have done experiments with different parameters, and we have noticed that a high reduction factor (which means low number of hidden units) is the most effective. We believe this is due to possible overfitting issues affecting the models that have more hidden units.
>
> - Number of hidden layers. We have tried 1, 2, and 3 hidden layers. Best results were obtained with 1 and 2 hidden layers. We think that increasing the number of hidden layers is deleterious because it can lead to overfitting issues.
>
> - Use of BatchNorm (BN). The use of BN is not possible in CaSE because the MLP takes a single data-point as input and not a mini-batch. This is due to the context aggregation, that produces a single aggregated vector which cannot be used to gather the statistics for the BN. Other normalization mechanisms could potentially be used but we did not explore this factor in our ablations.
>
> We will include these details and tables with numerical results in the revised version of the paper.

---

### Official Review · Reviewer_7nRY · 2022-07-11

**Rating:** 6
**Confidence:** 4
**Soundness:** 4 excellent
**Presentation:** 2 fair
**Contribution:** 3 good

**Summary:**

Authors correctly point out the tradeoff between adaptation costs and performance on novel datasets. Motivated by this tradeoff, authors introduce a novel adaptation layer, based on squeeze-and-excitation, that performs task-based feature modulation and is meta-learned in combination with a per-task learned linear head. The proposed method performs well on a variety of large-scale and difficult few-shot adaptation benchmarks at a reasonable computation cost.

**Questions:**

Given CaSE’s conceptual similarity to FiLM and TADAM, which I consider to be far more appropriate baselines than SE, what exactly do the authors consider to be the true contribution and proper source of novelty here?

Given the above answer, how can this position be strengthened and made clearer in the body text of the paper?


**Limitations:**

Limitations are discussed clearly and fairly. Societal impacts are addressed very briefly, though since the impacts match those of few-shot learning more broadly, this is sensible.

**Strengths And Weaknesses:**

STRENGTHS:

A well-motivated, well-reasoned and widely applicable approach. Results are convincing and the model clearly accomplishes what it sets out to accomplish, in the way that it claims to accomplish it. Paper is clearly written, if not entirely well-organized or -focused (see below).

WEAKNESSES:

The paper suffers from a misplaced focus in its presentation. Many of the named concepts presented as novel are not, and the truly novel contribution is somewhat limited and discussed only briefly.

1. CaSE is introduced as a novel layer but is identical to SE but for the context pooling (discussed below) and the final activation layer (an implementation detail discussed in supplementary).

2. Related to above, the adaptive mode / inference mode (described as “fundamental” on pg4 line 142) of CaSE is identical to TADAM (TADAM: Task Dependent Adaptive Metric for Improved Few-Shot Learning, NeurIPS2018), and also just a common-sense approach to handling a support vs query set.

3. I hesitate to call the UpperCaSE meta-training scheme particularly novel or coordinate-descent-based. While optimization does switch back and forth between CaSE and head parameters, the head parameters reset with every new batch. In actuality the CaSE parameters are being meta-learned, while head parameters are being set in the inner loop, and UpperCaSE is just a straightforward and common-sense approach to meta-training. This procedure (train linear head to convergence, propagate gradient into meta-learned layers) is also the exact same procedure already introduced by MetaOptNet (Meta-Learning with Differentiable Convex Optimization, CVPR2019).

In my eyes the true novelty of CaSE is in taking the FiLM/TADAM approach to task-conditioning, and replacing the redundant task encoder layers with the appropriate intermediate network activations. This crucial difference is discussed only briefly in related work (pg6 line 237-238). I also consider this a somewhat limited conceptual contribution, empirical results aside.

In my eyes, this calls for a fairly substantial text revision, where the contribution is mainly a novel _approach_ to efficient adaptation rather than a “new adaptive block” (lines 7-8), and the conceptual linkages to TADAM/FiLM are explored and discussed rather than the similarities/differences relative to SE, which are much less relevant in this context. I recognize this could be a pretty idiosyncratic and overly specific take though – I’ll be curious to see what other reviewers think on the novelty/contribution.

Less importantly, CaSE is conceptually quite similar to TADAM, and above proposed revisions aside, TADAM is at least worth a mention in related work, as a FiLM-derivative approach to few-shot learning.

---

> ### Author Response · Authors · 2022-08-01
> **Answer to reviewer 7nRY (part 1)**
>
> Thank you for the detailed comments on our paper, we appreciate your feedback which has which has raised numerous useful issues and insights.
>
> **1. “CaSE is introduced as a novel layer but is identical to SE but for the context pooling (discussed below) and the final activation layer (an implementation detail discussed in supplementary)”**
>
> We believe that CaSE and SE are qualitatively distinct as there are two major differences between them, and both contribute to the high-level performance of CaSE. The first crucial difference is the context pooling (as mentioned by the reviewer), SE is an instance-based method while CaSE is a context-based method. This difference is extremely important as we show in the experiments in Section 5 (Figure 3) where SE underperforms w.r.t. CaSE. The second difference is the fact that CaSE estimates the scale parameter on the context and then applies it on the target without additional forward passes on the MLPs. On the contrary, SE always performs a forward pass on the MLPs since there is no distinction between context and target. We agree with the reviewer that this is like the mechanism used in FiLM adapters (e.g. TADAM, SimpleCNAPs, LITE), but it was important to mention it in the paper since this mechanism is not used in standard SE.
>
> The reviewer has also mentioned the activation of the final layer in CaSE. The multiplier used in the activation function of the output layer in the MLPs can be considered a minor technical difference, therefore we have included it in the supplementary and not in the main text.
>
> We think that these distinctions make CaSE and SE fundamentally different since they are designed to work in completely different settings (standard supervised learning for SE vs. meta-learning for CaSE).
>
> These differences were discussed in Section 2 (and graphically in Figure 2) and an empirical comparison provided in Section 5 (Figure 3), we will improve the text to make this clear.
>
> **2. “[…] the adaptive mode / inference mode (described as “fundamental” on pg4 line 142) of CaSE is identical to TADAM (TADAM: Task Dependent Adaptive Metric for Improved Few-Shot Learning, NeurIPS2018), and also just a common-sense approach to handling a support vs query set”**
>
> We agree with the reviewer, and we will clarify. There has been a misunderstanding regarding the word “fundamental”, we meant that the use of adaptive/inference mode is crucial for obtaining a high accuracy w.r.t. the SE baseline that does not use it (as showed in the experiments in Section 5.1). We are aware of the fact that similar mechanisms are used in other adapters (e.g. FiLM adapters used in TADAM, SimpleCNAPS, and LITE). This will be clarified in the paper.
>
> **3. “UpperCaSE is just a straightforward and common-sense approach to meta-training. This procedure (train linear head to convergence, propagate gradient into meta-learned layers) is also the exact same procedure already introduced by MetaOptNet”**
>
> We agree with the reviewer, UpperCaSE is a straightforward and common-sense approach to meta-learning. We consider this to be an advantage of the method. More complicated approaches, such as implicit gradients (see [4][5]) have been far more popular. In our paper, we have shown that this simple technique can be effective in large-scale few-shot learning problems.
>
> Regarding the comparison with MetaOptNet, we were not aware of this paper. We have checked it and we acknowledge that UpperCaSE is similar in spirit to MetaOptNet in the way head and body are alternatively optimized. This part will be amended in the paper. However, while UpperCaSE is not novel in using an alternating optimization, it is novel in the way implements, manages, and updates the head in the inner loop, which is significantly different and more efficient than MetaOpNet.
>
>  MetaOpNet relies on SVMs for the final layer which can be problematic for large tasks due to the use of a quadratic solver. This is discussed in the MetaOpNet paper, at the end of Section 3.3. where the authors describe that the time complexity is cubic in the number of optimization variables (the dual formulation partially solves the issue when the number of training examples and the number of classes is small). This can be a significant obstacle for MetaOptNet, since in modern meta-learning benchmarks such as MetaDataset and VTAB the tasks are typically large (e.g. thousands of points). Our method (UpperCaSE) does not have the same issue, since the last layer is a standard fully-connected layer and the minimum is reached with an efficient stochastic optimization routine such as SGD. UpperCaSE exploits mini-batching which means that the time complexity is constant w.r.t. the size of the task.
>
> We will acknowledge the MetaOptNet paper and describe the similarities/differences with UpperCaSE in the revised version of the paper (Section 3).

---

> > ### Author Response · Authors · 2022-08-01
> > **Answer to reviewer 7nRY (part 2)**
> >
> > **4. “[…] the true novelty of CaSE is in taking the FiLM/TADAM approach to task-conditioning, and replacing the redundant task encoder layers with the appropriate intermediate network activations.”**
> >
> > We agree with the reviewer that this is a core contribution of our paper. We also agree that this point was not highlighted properly in the paper. In the revised version we will include an additional sub-section under Section 2 where we will discuss in detail the differences between CaSE and the FiLM adapters used in TADAM, SimpleCNAPS, and LITE. The bullet points for the main contributions in Section 1 will also be updated.
> >
> >
> > **5. “[…] the contribution is mainly a novel approach to efficient adaptation rather than a “new adaptive block” (lines 7-8), and the conceptual linkages to TADAM/FiLM are explored and discussed rather than the similarities/differences relative to SE, which are much less relevant in this context.”**
> >
> > We can make this part more explicit in the revised version by modifying the bullet points of the main contributions at the end of Section 1. The relationship between CaSE and other adaptation mechanisms (TADAM, SimpleCNAPS, LITE) will be discussed thoroughly in a dedicated sub-section (in Section 2). We still think that the comparison with standard SE is important as CaSE builds on top of it, therefore the reader can benefit from the discussion provided in Section 2 and Figure 2.
> >
> >
> > **6. “Less importantly, CaSE is conceptually quite similar to TADAM, and above proposed revisions aside, TADAM is at least worth a mention in related work, as a FiLM-derivative approach to few-shot learning.”**
> >
> > We thank the reviewer for pointing out this paper. We agree that TADAM should be mentioned and discussed. We would like to point out that the adaptation mechanism used in TADAM is like the one used in Simple-CNAPs [1] and LITE [2]. Simple-CNAPs has been shown to be superior to TADAM in [1], and LITE to be superior to SimpleCNAPs in [2]. Therefore, in the paper we have empirically compared against LITE, which is the most recent and best performing method overall.
> >
> > There are important differences between CaSE and the adaptation mechanism used in TADAM, SimpleCNAPs, and LITE (based on FiLM). The advantage of CaSE is that it does not need a dedicated set-encoder (as pointed out by the reviewer), therefore it is significantly more efficient in terms of parameters and more effective in terms of accuracy. Another difference is that CaSE only uses a scale vector to modify the feature maps, whereas FiLM uses both a scale and a shift vector (twice the number of parameters).
> >
> > These two differences make CaSE qualitatively better than FiLM adapters in terms of efficiency and superior in terms of accuracy as we showed in the experiments. Moreover, from a practical point of view CaSE is much easier to implement and use than the FiLM adapters used in TADAM, SimpleCNAPs, and LITE, since it can be plugged between the layers of a backbone (e.g. a single line of code in a Pytorch sequential module) without the overhead implicit in managing a set-encoder.
> >
> > We have compared CaSE and the FiLM adapters used in LITE in Section 5 (Figure 3), showing that CaSE is superior in terms of accuracy. Moreover, the same methods have been compared in Section 5.2 (Table 2) in terms of number of adaptive parameters, where LITE requires 1.7M adaptive parameters while UpperCaSE only 0.4M.
> >
> > Overall, we agree with the reviewer that these differences were not discussed adequately in the paper. We will make this more explicit in the revised text by adding a sub-section under Section 2 where we will compare CaSE to the adaptive mechanisms used in TADAM, SimpleCNAPs, and LITE.

---

> > > ### Author Response · Authors · 2022-08-01
> > > **Answer to reviewer 7nRY (part 3)**
> > >
> > > **7. “Given CaSE’s conceptual similarity to FiLM and TADAM, which I consider to be far more appropriate baselines than SE, what exactly do the authors consider to be the true contribution and proper source of novelty here?”**
> > >
> > > We believe that our paper offers different novel contributions and take-home lessons that can be of interest for the community. Here we provide a list of what we think are the main elements of novelty:
> > >
> > > - CaSE, a novel approach to efficient adaptation which is superior to concurrent methods (e.g. FiLM used in TADAM, SimpleCNAPs, LITE) in terms of accuracy and number of adaptive parameters. CaSE is easy to use and implement, representing an effective alternative to the widely used FiLM adapters.
> > >
> > > - UpperCaSE, a Coordinate-Descent approach based on the alternating optimization of head and body (similar to MetaOpNet) which exploits a fully-connected layer in the head to efficiently adapt to few-shot learning tasks.
> > >
> > > - State-of-the-art results on challenging few-shot learning datasets (MD, VTAB, ORBIT) with large images and backbones, which highlight how our method is the best trade-off in terms of accuracy and adaptation cost, making it the optimal solution for real-world applications such as personalization.
> > >
> > > - Experimental comparison of adaptation mechanisms (Section 5.1., Figure 3). We consider these experiments important in terms of novelty, because the FiLM adapters used in TADAM, SimpleCNAPs, and LITE have been widely used by the community and taken for granted. In our paper we have showed that it is possible to build more efficient adapters such as CaSE by reconsidering some of the assumptions of FiLM adapters (e.g. the need for scale and shift and the need for a set-encoder).
> > >
> > > **8. “Given the above answer, how can this position be strengthened and made clearer in the body text of the paper?”**
> > >
> > > We propose a set of changes to improve the clarity of the paper and the description of the main contributions:
> > >
> > > - In Section 2, where we introduce CaSE, we will include a dedicated sub-section where we will compare CaSE to the FiLM adapters used in TADAM, SimpleCNAPs, and LITE. As the reviewer pointed out, in the original version of the paper this was only briefly described in the “Related Work” section. We believe that having a dedicated sub-section with the details discussed in the rebuttal will improve the manuscript.
> > >
> > > - In Section 3, where we introduce UpperCaSE and the Coordinate-Descent algorithm we will mention the MetaOpNet paper and discuss the similarity/differences between the two solutions.
> > >
> > > - We will revise the main contributions in the Introduction (Section 1) to reflect these changes.
> > >
> > >
> > >
> > > **References**
> > >
> > > [1] Bateni, P., Goyal, R., Masrani, V., Wood, F., & Sigal, L. (2020). Improved few-shot visual classification. In Proceedings of the IEEE/CVF Conference on Computer Vision and Pattern Recognition (pp. 14493-14502).
> > >
> > > [2] Bronskill, J., Massiceti, D., Patacchiola, M., Hofmann, K., Nowozin, S., & Turner, R. (2021). Memory efficient meta-learning with large images. Advances in Neural Information Processing Systems, 34, 24327-24339.
> > >
> > > [3] Bronskill, J., Gordon, J., Requeima, J., Nowozin, S., & Turner, R. (2020, November). Tasknorm: Rethinking batch normalization for meta-learning. In International Conference on Machine Learning (pp. 1153-1164). PMLR.
> > >
> > > [4] Rajeswaran, A., Finn, C., Kakade, S. M., & Levine, S. (2019). Meta-learning with implicit gradients. Advances in neural information processing systems, 32.
> > >
> > > [5] Chen, Y., Friesen, A. L., Behbahani, F., Doucet, A., Budden, D., Hoffman, M., & de Freitas, N. (2020). Modular meta-learning with shrinkage. Advances in Neural Information Processing Systems, 33, 2858-2869.

---

> ### Comment · Reviewer_7nRY · 2022-08-06
> **Response to authors and clarification**
>
> Thank you for the detailed response and useful clarifications, and apologies for any lack of clarity and my poor response time, as I’ve been traveling this week. It seems I share my concern with reviewer FFac on better contextualization with prior work, but authors’ revisions have addressed many issues here (though with one glaring exception, see next pgraph). For the record, I agree that CaSE is interesting, effective, and fairly novel; the fact that it is a straightforward application of common-sense principles is a strength of the method, though slightly damaging in terms of conceptual novelty, and therefore intellectual contribution – I view this as a tradeoff rather than a weakness. My foremost issue was in presentation.
>
> My main outstanding concern (and apologies again for my failure to elaborate properly in my initial review) is the UpperCaSE training scheme, which I simply do not consider to be novel. While I cited MetaOptNet as support for this, I had intended to use it as just an example of inner-loop-head/outer-loop-body meta-training as a broadly pre-existing and accepted technique, not as a direct comparison. Other examples, and perhaps more appropriate comparisons, would be R2D2/LRD2 (Meta-Learning with Differentiable Closed-Form Solvers, ICLR19) which like MetaOptNet solves for the linear head in closed form, and more directly, ANIL (Rapid Learning or Feature Reuse? Towards Understanding the Effectiveness of MAML, ICLR20), which optimizes just the head in the inner loop using SGD, and then propagates gradients into the body for meta-training, identical to UpperCaSE. Thus I do not believe that UpperCaSE can be claimed as a separate contribution, as this style of meta-training already exists in multiple instances and incarnations in prior literature. For this reason I have not yet updated my rating – apologies again for my initial lack of clarity here.

---

> > ### Author Response · Authors · 2022-08-07
> > **Answer to Reviewer 7nRY**
> >
> > Thank you for engaging in the discussion and for providing additional information on your previous comments.
> >
> > **1. [...] “authors’ revisions have addressed many issues here (though with one glaring exception, see next pgraph). For the record, I agree that CaSE is interesting, effective, and fairly novel; the fact that it is a straightforward application of common-sense principles is a strength of the method, though slightly damaging in terms of conceptual novelty, and therefore intellectual contribution – I view this as a tradeoff rather than a weakness.”**
> >
> > We are glad that the rebuttal has clarified these points and that the reviewer’s view is aligned with our own on the core contribution of the paper.
> >
> > **2. “[...] Thus I do not believe that UpperCaSE can be claimed as a separate contribution, as this style of meta-training already exists in multiple instances and incarnations in prior literature.”**
> >
> > Thank you for clarifying this point and providing more details regarding this issue. We now have fully understood the reviewer’s concern. By looking at the evidence provided by the reviewer we agree that UpperCaSE cannot be claimed as a novel contribution.
> >
> > To amend this issue, we propose the following updates to the paper:
> >
> > - We will modify the main contributions at the end of Section 1 removing point 2, the contribution that is mentioning UpperCaSE.
> > - We will update Section 3, where we introduce UpperCaSE, and mention that the method is a special case of a meta-training scheme that already exists. Here we will mention and discuss the papers cited by the reviewer (MetaOptNet, R2D2/LRD2, ANIL).
> > - The body of the document will be updated accordingly.
> >
> > Those changes will be added to a rebuttal version of the paper shortly.

---

> > > ### Comment · Reviewer_7nRY · 2022-08-08
> > > **Response to authors**
> > >
> > > My thanks to the authors for the response and update, authors' level of engagement and willingness to revise the submission on short notice is appreciated. The new revisions have mostly addressed my concerns, and I have updated my rating to Weak Accept.

---

> > > > ### Author Response · Authors · 2022-08-08
> > > > **Thank you**
> > > >
> > > > Thank you, we also appreciate the exceptional level of engagement shown by the reviewer during the rebuttal session.
> > > >
> > > > The reviewer has provided a very constructive feedback that has been fundamental in improving the paper.

---

### Author Response · Authors · 2022-08-01
**Answer to all reviewers**

We thank all the reviewers for their feedback on our paper which contributes:

- CaSE, an efficient adaptive-block similar to Squeeze-and-Excite that allows adjusting the body of a pretrained model on a given task with a single forward step.

- UpperCaSE, a procedure that exploits CaSE in the body of the network and a fine-tuning procedure in the linear head for rapid adaptation in the few-shot setting;

- A wide range of experiments consisting of: (i) comparison on challenging few-shot learning benchmarks (MD, VTAB, ORBIT) with state-of-the-art accuracy for meta-learners; (ii) an empirical comparison of adaptation mechanisms (SE, CaSE, FiLM) showing the superior efficiency and accuracy of CaSE.

All the reviewers agreed that our work has provided an extensive set of experiments showing that the proposed method is “well-motivated, well-reasoned and widely applicable” (Reviewer 7nRY), “simple and novel yet achieves very good results compared to other meta-learners” (Reviewer 9nir), and “interesting and can be a potential plug-in solver for other meta-learning methods” (Reviewer FFac).

The reviewers primarily requested clarifications regarding the comparison with previous work and the contributions of the paper. We have answered individually to each reviewer. Here we provide a short summary and invite the reviewers to read the individual sessions for more detailed comments.

- Similarities with previous work (e.g. TADAM, MetaOpNet, MTL) and clarifications regarding elements of novelty (Reviewers 7nRY and FFac). We have discussed the similarities/differences with these papers in the rebuttal. We did not know some of these papers and we thank the reviewers for pointing them out. We acknowledge that there are similarities, but there are also some key differences that make our method novel and more efficient (efficiency is one of the main factors considered in our paper). This will be amended in the revised version. See answers to Reviewers 7nRY and FFac for more details.

- Ablation studies for the CaSE block (Reviewer 9nir). We have provided a detailed description of the ablations we have performed on the CaSE block. In the original manuscript we had reported only the ablation that had a significant impact on the results (e.g. the activation function in the output layer of CaSE). However, we also performed additional ablations on the number of hidden layers/neurons and activation functions that will be reported in the updated version of the paper. See answers to Reviewers 9nir for more details.

---

### Author Response · Authors · 2022-08-05
**Update to all reviewers**

We have uploaded a rebuttal version of the paper and supplementary, where the proposed changes have been highlighted in red. This is a summary of the changes:

1. Minor rewording of the main contributions in Section 1 (not marked to avoid cluttering).

2. New paragraph under Section 2 called “Comparison with other adapters” where we discuss the differences between the proposed CaSE and the FiLM adapters used in TADAM, CNAPs, SimpleCNAPS and LITE (as discussed in the answer to Reviewer 7nRY).

3. New paragraph in Section 3 (under the " Meta-training via Coordinate-Descent" sub-section) where we discuss the similarities/differences with MetaOptNet (as discussed in the answer to Reviewer 7nRY).

4. We have extended the paragraph called "Dynamic Networks" in Section 4 (Related work) to include and discuss the two papers pointed out by Reviewer FFac.

5. We have added the results of experiments concerning ablation studies in the supplementary material. In particular, we have created a new section called “Ablation studies” in Appendix C.4 as requested by Reviewer 9nir. The previous table reporting the ablation of the activation in the output layer has been moved under this new section.

We invite the reviewers to add any further comments if they have. We are happy to provide additional clarifications if needed.

---

### Author Response · Authors · 2022-08-07
**Second update to all reviewers**

We have modified the rebuttal version of the paper based on the considerations reported in the additional discussion with Reviewer 7nRY and Reviewer FFac. The proposed changes have been highlighted in red in the paper and the supplementary. This is a summary of the changes:

1. We have reviewed the main contributions in Section 1 removing the one related to UpperCaSE, as discussed with Reviewer 7nRY. This change has also been propagated to the abstract and part of the introduction.
2. We have modified Section 3, where UpperCaSE was introduced, to highlight the connection with previous work (R2D2, MetaOptNet, ANIL) as pointed out by Reviewer 7nRY.
3. We have included a revised discussion of the MTL paper (Section 4) based on the feedback given by Reviewer FFac.

We thank the reviewers for engaging in further discussion. We believe that the revised version of the paper is significantly clearer now.

---

### Meta-Review · Area_Chair_A8SR · 2022-08-29

**Recommendation:** Accept
**Confidence:** Certain

**Metareview:**

The reviewers consider the work technically solid, but were concerned about the contextualization of this work in the literature.  post-rebuttal, some of the reviewers concerns are resolved.

**Award:**

No

---

### Decision · Program_Chairs · 2022-09-14

Accept